# The Optimization between Physician Satisfaction and Hospital Profit in Cross-Hospital Scheduling—A Case Study of Some Hospitals in Taiwan

**DOI:** 10.3390/healthcare9081004

**Published:** 2021-08-05

**Authors:** Yi-Chao Huang, Jong-Ching Hwang, Yi-Chun Lin

**Affiliations:** 1Department of Industrial Management, National Pingtung University of Science and Technology, Neipu 91201, Taiwan; b10057089@yahoo.com.tw; 2Department of Electrical Engineering, National Kaohsiung University of Science and Technology, Kaohsiung 807618, Taiwan; ching@mail.ee.nkust.edu.tw

**Keywords:** cross-hospital scheduling, decision making, job satisfaction, integer programming, resource integration

## Abstract

In recent years, the majority of the population has preferred to go to large hospitals regardless of the severity of their illnesses, resulting in a waste of medical resources. In view of this situation, the government has proposed a cross-hospital integration plan to promote the integration of medical resources. Hospitals that provide support can not only increase their income but also extend their medical coverage to other regions and get wide access to more patients. While previous studies mainly focused on the internal shift scheduling of hospitals, this study took into account both the internal hospital and the support to branch hospitals and particularly explored the financial benefits generated by the provision of support and the satisfaction of physicians on shift scheduling. Decision makers can give different weight values according to the management needs and then determine the most appropriate physician shift schedule according to the final decision value. The shift-schedule-building system developed in this study could be used to quickly calculate the most appropriate shift schedule according to the actual needs, which could replace the time-consuming method of manual scheduling and improve physician satisfaction and hospital financial income.

## 1. Introduction

Taiwan implemented the national health insurance system on 1 March 1995 to provide medical and health care services to its people. The system has contributed to the country’s medical development and realized the concept of public sharing and equal access to medical care. However, this has also led to a substantial increase in medical expenditures that is far beyond the scope of the originally approved premium. To solve this problem, the National Health Insurance Administration has introduced a global payment system to control its financial budget. In order to foster the medical classification, the government encourages hospitals to cooperate at different levels to provide integrated care. Meanwhile, the government has also implemented a referral system through the national health insurance to guide patients to consult grass-roots hospitals for treatments on small illnesses and large hospitals for serious illnesses so as to achieve the objective of reducing the waste of medical resources. Cross-level cooperation can not only promote the integration of the medical system and reduce the competition between hospitals but also promote the physicians of large- and medium-sized hospitals to support community hospitals and improve the quality of medical services in the communities so that medical resources can be reasonably and effectively distributed and allow the public to visit hospitals nearby and enjoy medical services of the same quality.

Through physician support, both large and community hospitals can achieve mutual benefit. On the one hand, a physician’s main hospital can earn additional income from the physician’s support to branch hospitals; on the other hand, branch hospitals can improve their medical quality and thus attract more patients, which can increase their profits. However, under this mode, physicians will have to provide outpatient services in two or more hospitals. Without a proper shift schedule, the physicians may have to work shifts in different hospitals on the same day, which will cause the physicians to suffer from fatigue due travel or be unable to catch up with the service time, thus affecting the rights and interests of patients. Therefore, in the actual scheduling of hospitals, decision makers should not only consider a single objective but also take into account of other objectives or constraints, such as hospital profits and physician satisfaction. However, increasing the hospital income from providing support to branch hospitals will affect the physicians’ satisfaction with the shift schedule. Physicians with senior qualifications (who have served more than 15 years) and rich experiences can bring more income to their own hospitals by providing support to branch hospitals, but more senior physicians will be arranged to serve in the same hospital from the perspective of medical ethics. The above situation will increase the complexity of scheduling. Therefore, this study adopted the integer programming method to solve the problem of multi-objective scheduling. On the one hand, a computer was employed to replace traditional manual scheduling and improve the efficiency of scheduling. On the other hand, a fair and reasonable shift schedule for physicians was developed to achieve the objectives of maximizing hospital income and physician satisfaction.

The way of physician support will affect the physician-scheduling results under different objectives. The advantage is that it enables the hospital to receive additional income and improve profits, while the disadvantage is that it will cause the physicians to work in different hospitals, thus making them suffer from fatigue due to travel. The purpose of this study was to adopt the integer programming method to explore the impact of physicians’ schedules on the financial revenue of the hospital and the physician satisfaction with the scheduling as well as to find the most satisfactory solution that meets both objectives at the same time so as to provide decision makers with an optimal shift schedule.

## 2. Literature Review

### 2.1. Linear Programming

Linear programming is used to construct a mathematical model through the development of objectives and constraints, while the integer programming method in operations research is used to solve linear or nonlinear problems of the integer value of decision variables. Generally, integer programming is regarded as a special case of linear programming that is applied to solve many practical problems, such as assignment, location, transportation, and logistics problems, as well as shift and course scheduling. In the field of transportation and logistics, distribution mostly takes minimizing storage and transportation costs as the goal. By using the integer programming method, it is possible to accurately calculate the optimized distribution route and warehouse location [1] or arrange the optimal number of vehicle trips to reduce operation costs [2]. Therefore, the integer programming model is helpful in supply-chain-network management because it considers the combination of each manufacturing plant, production capacity, transportation, and other links [3,4] so as to optimize the network configuration and reduce operating costs. Effective personnel arrangement has become the most competitive method in service entities. In the United States, the postal-service industry has applied pure integer linear programming to achieve the goal of minimizing personnel costs per day without violating labor contracts or government regulations [5]. The complexity of distribution is just like the scheduling of shifts between hospitals, which involves multiple clinics and objectives. As different types of medical clinics will have different personnel needs, integer programming can be used to achieve the optimal distribution of personnel according to people’s preferences and after taking into account the distance between people and clinics, the time, and the service demand [6]. Thus, integer programming is not only used in the field of transportation and distribution but is also popularly applied to personnel scheduling. In the aviation industry, Abara applied the integer linear programming model to realize the assignment of two or more crews to a single flight schedule [7]. The most important goal of airline crew matching is to minimize the expenses related to the assignment. For example, in order to reduce operating costs, Air France uses the integer nonlinear multi-commodity network flow model to generate feasible flights and uses the branch-and-bound method to find solutions that will reduce operating costs [8].

### 2.2. Scheduling Issues

Personnel scheduling usually deals with multi-task assignments that must satisfy some constraints on multiple personnel at the same time. For example, by using the 0–1 linear programming, Azaiez and Sharif found out that with appropriate personnel, nursing skills can be reasonably applied to consecutive services and unnecessary additional costs can be avoided [9]. For example, overtime work can be arranged to meet the hospital’s objectives. Therefore, integer programming has an extensive scope of application. It can be combined with systematization or various algorithms to improve the execution efficiency. Its purpose is to optimize the combination of problems, meet objectives and various constraints, achieve appropriate assignment and planning, and reduce the planning time [10]. Personnel scheduling can achieve the effective management of human resources by arranging appropriate personnel for the required work at the right time so that the needs and objectives of the unit can be met [11]. Meisels and Schaerf defined scheduling as an effective shift combination that has personnel assigned in different shifts, with each shift usually based on a fixed time [12]. Many scheduling cases are needed to meet work needs with minimal cost or maximum benefits [13,14]. With the improvement of workers’ awareness, scheduling has gradually turned to the construction of a complete and proper shift schedule by taking into account factors with multiple natures, such as rationality and fairness, as well as the expectations of employees, rotating shifts, and vacation time [15,16]. Different industries pursue the reasonability and efficiency of work by improving the scheduling mode. The aviation company constructed the mathematical model [17] of flight attendants’ scheduling through integral programming and applied sample average approximation (SAA) algorithm to solve the scheduling issue of ballooning [18]. The integral programming could also be used to solve the dispatching of bus shifts and the meal problems of drivers [19,20]. The scheduling of policemen belongs to the scheduling issue of ordinary personnel. Besides the discussion of the patrol area [21], the questionnaire was adopted to understand different demands of each policeman and formulate the service schedule that meets various demands of the personnel [22].

In the medical field, the actual work and staffing of physicians is quite complex because they need to participate in informal cooperation and support in case of an emergency [23,24]. As a result, it is difficult to make general rules. Doctors generally have personal and special contractual obligations. Therefore, most doctors’ schedules only consider doctor satisfaction. Doctors at different positions in various departments will have different types of work and must face the various work-performance indicators from the management departments [25]. Therefore, a good schedule is very important in medical institutions. If there is no proper arrangement, it may cause the interval and delay of medical service times or the cancellation of operations, thus reflecting the excess cost of the hospital and leading to the loss of the hospital [26]. 

Nowadays, patients have increasingly higher demands for medical services, which leads to increasingly heavier workloads and pressure for medical staff. Therefore, it is important to arrange a reasonable division of labor, emphasize the diversity of work, and meet the reasonable needs of personnel. However, in most medical institutions, the objectives of medical staff scheduling are mainly to ensure continuous services and to meet the scheduling strategy of the organization while at the same time using minimum manpower allocation to avoid the waste of manpower, which can reduce unnecessary overtime work and other additional expenses and thus reduce the labor cost of the hospital [9]. Beaulieu et al. made a shift schedule for physicians in an emergency room using the multi-objective integer programming model, which could accommodate such rules as seniority and personal preferences [27]. However, scheduling needs to consider many factors, making it an NP-Hard issue [28]. Reasonable scheduling and work rotation can improve work performance and reduce employee fatigue and work injury [29]. Therefore, a shift schedule that can evenly share the fatigue and meet the holiday preferences of each employee can achieve the goal of reducing the maximum fatigue value. In addition, the quality of medical staff scheduling also involves the level of labor cost and further affects the operation level and quality of patient care [30].

Mathematical programming methods are not the only solution to hospital manpower scheduling. For example, Ferrand et al. constructed an eight-week shift schedule system for physicians-in-charge through linear programming [31]. Different methods of hospital manpower scheduling have been documented in many literatures. Puente et al. used the genetic algorithm to solve the average shift schedule of emergency room medical staff in 24-h shifts [32]. Tsai and Li applied the genetic algorithm to the two-stage mathematical model and combined it with interactive web technology to effectively solve the shift schedule of medical staff and provide more respect for self-determined schedules, which could increase the staff’s satisfaction [33]. Finally, Topaloglu and Selim applied fuzzy theory to deal with the uncertainty between the target value of hospital management and nurses’ preferences for duty [34]. 

According to the above literature, it could be found that it is common to apply integer programming to personnel scheduling in the medical system. At present, most scheduling is based on preferences or to minimize the situation of redundant or insufficient personnel so as to reduce personnel costs and arrange a reasonable schedule. The above literatures on scheduling only focused on single goals, such as preferences or the fair number of vacations and shifts, but they did not take into account the need for personnel scheduling in multiple hospitals or the impact of the provision of medical services by personnel with different qualifications in different hospitals on hospital revenue, all of which are related to a hospital’s operating objectives or operation mode. Therefore, this study focused on the rationality of scheduling to meet the scheduling objectives for hospital revenue and physician satisfaction so as to provide decision makers with an optimal physician schedule.

## 3. Research Method

### 3.1. Definition of Variables


**Name of set**

*J*
Set of physicians
*K*
Set of weekdays
*L*
Set of shifts
*M*
Set of outpatient rooms
*N*
Set of hospitals
**Parameter**

*j*
Code of physician: senior physician, *j* = 1,2,…,N_a_; junior physician, j = N_a_ + 1, N_a_ + 2,...,***J***
*k*
Week: *k* = 1,2,…,*K* (*k* = 1, Monday; *k* = 2, Tuesday…, *k* = 6, Saturday) 
*l*
Time: *l* = 1,2,…,***L*** (*l* = 1, morning shift; *l* = 2, afternoon shift; *l* = 3, evening shift) 
*m*
medical service: *m* = 1,2,…,*M*; *M* = max (N_m1_, N_m2_, N_m3_) 
*n*
hospital: *n* = 1,2,…,*N* (*n* = 1, main hospital; *n* = 2, branch; *n* = 3, nursing home)
**Definition of symbols**

*PZ_n_*
The maximum income under scheduling scenario *n*
*PRW_n_*
The relative weight of the income under scheduling scenario *n*

*S_a_*
Physician satisfaction with scheduling of inconsecutive shifts in the same hospital
*S_b_*
Physician satisfaction with scheduling of consecutive shifts in the same hospital
*S_c_*
Physician satisfaction with scheduling of inconsecutive shifts in different hospitals 
*SZ_n_*
Maximum satisfaction under scheduling scenario *n*
*SRW_n_*
Relative weight of maximum satisfaction under scheduling scenario *n*
*DV_n_*
Decision value of scheduling under scenario *n*
*DW*
Decision weight value
*MDV_n_*
Maximum decision value
*N_w_*
Outpatient service demand in one week
*M_j_*
Physician *j*’s outpatient number in one week
*D_jk_*
Physician *j* needs to make outpatient surgery clinic on *k* (day of the week)
*N_d*1*_*
Minimum number of shifts in one day
*N_d*2*_*
Maximum number of shifts in one day
*N_m*1*_*
Number of outpatient rooms in the main hospital
*N_m*2*_*
Number of outpatient rooms in the branch
*N_m*3*_*
Number of outpatient rooms in the nursing home 
*N_s*1*_*
Total number of senior physicians who support the branches and nursing homes
*N_s*2*_*
Total number of junior physicians who support the branches and nursing homes
**Decision Variable**

*X_jklmn_*
Whether physician *j* takes the shift *l* of outpatient room *m* in hospital *n* on day *k* (0 indicates “no”; 1 indicates “yes”) 
*Y_jk_*
Whether physician *j* works on day *k* (0 indicates “no”; 1 indicates “yes”)

### 3.2. Description of Physician Scheduling Problem

In order to integrate regional medical resources, hospitals of different scales will take a business model of cross-hospital integration. Large hospitals will assign physicians to smaller hospitals to support their medical work, which can not only increase the medical level of the branch hospitals but also increase the number of patients in the main hospital through a referral system. Providing support to branch hospitals can increase the income of the main hospital. However, because physicians have to serve across hospitals, when there is a lack of transportation or medical equipment, management must consider physician satisfaction with providing cross-hospital services.

When experienced physicians stay in branch hospitals to support medical services, they can attract more patients to visit the supported hospitals; therefore, the supported hospitals are willing to hire senior physicians at a high cost. The number of senior or junior physicians assigned will affect the income of the supporting hospitals. According to the questionnaire survey, the ranking of physician satisfaction on three types of scheduling (from high to low) was: (1) inconsecutive afternoon shifts in the same hospital; (2) consecutive afternoon shifts in the same hospital; and (3) inconsecutive afternoon shifts in different hospitals. However, due to the traffic problems in different hospitals, scheduling consecutive afternoon shifts in different hospitals and similar scheduling strategies are also avoided. In order to respect workplace ethics, management can give a higher weight value of satisfaction to senior physicians. Other management constraints are explained in the constraint of linear programming in this paper. This study constructed a 0–1 integer programming model that could allow management to plan the cross-hospital scheduling of physicians according to prioritization between medical service income and satisfaction. This scheduling model could help management make the most appropriate scheduling that balances support income (income from providing supporting services) with satisfaction within the shortest time.

### 3.3. Construction and Description of the Physician-Scheduling Model

The scheduling model of this study was constructed through 0–1 integer programming based on hospital support income and physician satisfaction. Managers could use the model to set the relative weight of support income and satisfaction according to the practical needs and then find out the most appropriate physician schedule in the shortest time. 

#### 3.3.1. Physician satisfaction with the scheduling model

Objective function: Maximum satisfaction
Max *Z* = *S_a_* + *S_b_* + *S_c_*(1)

Equation (1) is the objective function of physician satisfaction with scheduling. This function consisted of three scheduling scenarios, including inconsecutive afternoon shifts in the same hospital, consecutive afternoon shifts in the same hospital, and inconsecutive afternoon shifts in different hospitals, which were multiplied by the satisfaction of the relative weight. 

To calculate the weight value, this study adopted the SMART-ROC equation, as shown in Equation (2):(2)$$W_{i}=\frac{1}{n}\sum_{k=i}^{n}\frac{1}{k}$$
where *n* indicates the number of attributes and *k* indicates the rank order of preference. 

Using the SMART-ROC method, physician satisfaction with scheduling was investigated and ranked, and the weight value was calculated according to the rank of the physician satisfaction. The obtained weight value was multiplied by 100 and rounded off to the integer part for convenience of calculation. In this study, the rank order of satisfaction was most satisfied (three points), sub-satisfied (two points), and least satisfied (one point). The rank order of scheduling was *S_a_* (inconsecutive shifts in the same hospital), *S_b_* (consecutive shifts in the same hospital), and *S_c_* (inconsecutive shifts in different hospitals). The method to calculate the weight of satisfaction with scheduling was as shown in Equations (3)–(5):

Calculation of the weight of scheduling *S_a_*:(3)$$W_{1}=\frac{1}{3}\cdot \left(\frac{1}{1}+\frac{1}{2}+\frac{1}{3}\right)*100=61$$

Calculation of the weight of scheduling *S_b_*: (4)$$W_{2}=\frac{1}{3}\cdot \left(\frac{1}{2}+\frac{1}{3}\right)*100=28$$

Calculation of the weight of scheduling *S_c_*:(5)$$W_{3}=\frac{1}{3}\cdot \left(\frac{1}{3}\right)*100=11$$

The following section describes the mathematical patterns of various scheduling scenarios within the objective function. 

(1) Physician satisfaction with the scheduling scenario of inconsecutive shifts in the same hospital.

Equation (6) shows the total number of inconsecutive shifts of senior physicians and junior physicians in the same hospital, multiplied by the weight value of individual physicians: (6)$$S_{a}=122\cdot \sum_{j=1}^{Na}\sum_{k=1}^{K}[(\sum_{m=1}^{M}X_{jk1m1}+\sum_{m=1}^{M}X_{jk3m1})\#2,1,0]+61\cdot \sum_{j=Na+1}^{J}\sum_{k=1}^{K}[(\sum_{m=1}^{M}X_{jk1m1}+\sum_{m=1}^{M}X_{jk3m1})\#2,1,0]$$
where 

[(value)#2,1,0] means that when value = 2, [] = 1; otherwise, [] = 0.

During situations where physicians take excessive time performing a medical service, arranging such physicians to continue working will affect the rights and interests of the next patients or cause the physicians to be unable to rest, thus affecting the quality of the medical service. However, when arranging the schedule, it is necessary to consider the perspective of medical ethics. Therefore, in the case of inconsecutive shifts in the same hospital, senior physicians were given twice the weight value of junior physicians, i.e., 122, while the weight value of junior physicians was 61, so that priority would be given to the satisfaction of senior physicians with the scheduling scenario.

(2) Physician satisfaction with the scheduling scenario of consecutive shifts in the same hospital. 

Equation (7) is the total number of consecutive shifts of senior physicians and junior physicians in the same hospital, multiplied by the weight value of individual physicians.
(7)$$\begin{array}{l} S_{b}=56\cdot (\sum_{j=1}^{Na}\sum_{k=1}^{K}[(\sum_{m=1}^{M}\sum_{l=1}^{2}X_{jklm1})\#2,1,0]+\sum_{j=1}^{Na}\sum_{k=1}^{K}[(\sum_{m=1}^{M}\sum_{l=1}^{2}X_{jklm2})\#2,1,0]+\sum_{j=1}^{Na}\sum_{k=1}^{K}[(\sum_{m=1}^{M}\sum_{l=2}^{3}X_{jklm1})\#2,1,0])+\\ 28\cdot (\sum_{j=Na+1}^{J}\sum_{k=1}^{K}[(\sum_{m=1}^{M}\sum_{l=1}^{2}X_{jklm1})\#2,1,0]+\sum_{j=Na+1}^{J}\sum_{k=1}^{K}[(\sum_{m=1}^{M}\sum_{l=1}^{2}X_{jklm2})\#2,1,0]+\sum_{j=Na+1}^{J}\sum_{k=1}^{K}[(\sum_{m=1}^{M}\sum_{l=2}^{3}X_{jklm1})\#2,1,0]) \end{array}$$
where

[(value)#2,1,0] means that when value = 2, [] = 1; otherwise, [] = 0.

Due to the service times of the hospital, there may be cases where physicians need to take consecutive shifts. However, when arranging the schedule, it is necessary to consider the perspective of medical ethics. Therefore, in the case of consecutive shifts in the same hospital, senior physicians were given twice the weight value of junior physicians, i.e., 56, while the weight value of junior physicians was 28, so that priority would be given to the satisfaction of senior physicians with the scheduling scenario.

(3) Physician satisfaction with the scheduling scenario of inconsecutive shifts in different hospitals.

Equation (8) is the total number of inconsecutive shifts of senior physicians and junior physicians in different hospitals, multiplied by the weight value of individual physicians.
(8)$$\begin{array}{l} S_{c}=11\cdot (\sum_{j=1}^{Na}\sum_{k=1}^{K}[(\sum_{m=1}^{M}X_{jk3m1}+\sum_{m=1}^{M}X_{jk1m2})\#2,1,0]+\sum_{j=1}^{Na}\sum_{k=1}^{K}[(\sum_{m=1}^{M}X_{jk3m1}+\sum_{m=1}^{M}X_{jk1m3})\#2,1,0])+\\ 22\cdot (\sum_{j=Na+1}^{J}\sum_{k=1}^{K}[(\sum_{m=1}^{M}X_{jk3m1}+\sum_{m=1}^{M}X_{jk1m2})\#2,1,0]+\sum_{j=Na+1}^{J}\sum_{k=1}^{K}[(\sum_{m=1}^{M}X_{jk3m1}+\sum_{m=1}^{M}X_{jk1m3})\#2,1,0]) \end{array}$$
where

[(value)#2,1,0] means that when value = 2, [] = 1; otherwise, [] = 0.

In view of physicians’ support of services in different hospitals and considering the inconvenience faced by physicians in providing services in branch hospitals or the possibility that they will be unable to catch up with consecutive service shifts, this scheduling mode was developed to ensure that physicians could have enough time to rest. As physicians are most unsatisfied with the scheduling scenario of shifts in different hospitals, this study gave lower weight values than those in the case of shifts in the same hospital. However, when arranging the schedule, it is necessary to consider the perspective of medical ethics. Therefore, in the case of inconsecutive shifts in different hospitals, junior physicians were given twice the weight value of senior physicians, i.e., 22, while the weight value of senior physicians was 11, so that priority would be given to the satisfaction of junior physicians with the scheduling scenario.

Constrains:(9)$$\sum_{k=1}^{K}\sum_{l=1}^{L}\sum_{m=1}^{M}\sum_{n=1}^{N}X_{jklmn}=M_{j}\;,\forall \;j$$

Constraint 9: Number of each physician’s shifts in one week. This formula represents that all outpatient volumes of each physician *j* in various hospitals within one week must be the same with the scheduling volume.
(10)$$\sum_{l=1}^{L}\sum_{m=1}^{M}\sum_{n=1}^{N}X_{jklmn}\leq 2\;,\forall \;j,k$$

Constraint 10: As the regular working hours shall be no more than eight hours per day, each physician can only take at most two shifts per day.
(11)$$\sum_{j=1}^{J}\sum_{k=1}^{K}\sum_{l=1}^{L}\sum_{m=1}^{M}\sum_{n=1}^{N}X_{jklmn}=Nw$$

Constraint 11: Total demand of each department in one week. This formula represents that the total sum of the outpatient volume of all physicians is the same as hospitals’ outpatient volume.
(12)$$N_{d1}\leq \sum_{j=1}^{J}\sum_{l=1}^{L}\sum_{m=1}^{M}\sum_{n=1}^{N}X_{jklmn}\leq N_{d2}\;,\;\forall \;k$$

Constraint 12: The number of shifts per day shall be between *N_d*1*_* and *N_d*2*_* so as to avoid the concentration of shifts on certain days. This formula represents that the daily outpatient volume of each physician must be within the stipulated outpatient volume. Generally, *N_d*2*_* cannot be over 2 for a single day, but it is not necessarily scheduled every day, so *N_d*1*_* is generally 0.
(13)$$\sum_{j=1}^{J}X_{jklmn}\leq 1\;,\;\forall \;k,l,m,n$$

Constraint 13: There can only be one serving physician within each service period.
(14)$$\sum_{m=1}^{M}\sum_{n=1}^{N}X_{jklmn}\leq 1\;,\;\forall \;j,k,l$$

Constraint 14: Each physician can only serve one shift within each period.
(15)$$\sum_{j=1}^{J}\sum_{m=1}^{M}\sum_{n=1}^{N}X_{jklmn}\geq 1\;,\;\forall \;k,l=1\sim 2$$

Constraint 15: In order to avoid the case in which any patient finds no physician in the hospital, outpatient service is required for every period; therefore, on each day, there must be at least one morning shift and one afternoon shift.
(16)$$\sum_{j=1}^{J}\sum_{m=1}^{M}\sum_{n=1}^{N}X_{jklmn}\geq 1\;,\;for\;l=3,k=1\sim 5$$

Constraint 16: The evening outpatient must be arranged per day in order to serve the patients that work in the daytime; outpatient service is required for every period; therefore, on each day, there must be at least one evening shift except for Saturdays.
(17)$$\sum_{j=1}^{J}\sum_{m=1}^{M}\sum_{n=1}^{N}X_{jklmn}=0,for\;k=6,l=3$$

Constraint 17: No outpatient is arranged on Saturday evenings, so no evening shift can be scheduled for physicians on Saturdays.
(18)$$Y_{jk}\geq X_{jklmn},\;\forall \;j,k,l,m,n$$

Constraint 18: If physicians have shifts in any hospital, the day’s schedule must have the schedule records; therefore, the relationship is established between X*_jklmn_* and Y*_jk_*.
(19)$$\sum_{k=1}^{k+2}Y_{jk}\leq 2\;,\;\forall \;j,k=1\sim 4$$

Constraint 19: To avoid consecutive shifts of physicians, each physician shall not provide services for more than two days consecutively.
(20)$$\sum_{j=1}^{J}\sum_{k=1}^{K}\sum_{m=1}^{N_{m2}}X_{jklmn}=0\;,for\;l=3,n=2$$

Constraint 20: As there is less demand for outpatient services in the branch hospitals, there are no night shifts in the branches.
(21)$$\sum_{j=1}^{J}\sum_{k=1}^{K}\sum_{l=1}^{L}\sum_{m=N_{m2}+1}^{M}X_{jklmn}=0\;,for\;n=2$$

Constraint 21: As there is less demand for outpatient service in the branch hospitals, the branch hospitals only have 1~*N_m*2*_* rooms to provide outpatient services, unlike the need for numerous outpatient rooms at the main hospital, so as to avoid the waste of resources.
(22)$$\sum_{j=1}^{J}\sum_{l=1}^{L}\sum_{m=1}^{N_{m}{}_{2}}X_{jklmn}=0\;,for\;k=6,n=2$$

Constraint 22: The outpatient volume in branch hospitals is small on Saturdays; hence, no shifts can be scheduled in the branch hospitals on Saturdays.
(23)$$\sum_{j=1}^{J}\sum_{m=1}^{N_{m2}}X_{jklmn}\geq 1\;,\;for\;k=1\sim 5,\;l=1,2,n=2$$

Constraint 23: Branch hospitals have no evening outpatients; therefore, the branch hospital must have morning and afternoon shifts scheduled every day.
(24)$$\sum_{j=1}^{J}\sum_{k=1}^{K}\sum_{l=1}^{L}\sum_{m=N_{m3}+1}^{M}X_{jklmn}=0,for\;n=3$$

Constraint 24: Nursing homes only provide services to the elderly; therefore, they may only have 1~*N_m*3*_* rooms to provide outpatient services, unlike the need for numerous outpatient rooms at the main hospital, so as to avoid waste of resources.
(25)$$\sum_{j=1}^{J}\sum_{k=1}^{K}\sum_{l=2}^{3}\sum_{m=1}^{N_{m3}}X_{jklmn}=0,for\;n=3$$

Constraint 25: Nursing homes usually receive fewer visits, so they may not schedule afternoon and evening shifts.
(26)$$\sum_{j=1}^{J}\sum_{m=1}^{N_{m3}}X_{jklmn}=1\;,\;for\;k=1,2,4\;,\;l=1,\;n=3$$

Constraint 26: Due to a smaller outpatient volume, nursing homes can schedule morning shifts on Mondays, Tuesdays, and Thursdays.
(27)$$\sum_{j=1}^{J}\sum_{l=1}^{L}\sum_{m=1}^{N_{m3}}X_{jklmn}=0\;,for\;k=3,5,6\;,\;n=3$$

Constraint 27: As there is less demand for outpatient services in nursing homes, they may not schedule shifts on Wednesdays, Fridays, and Saturdays.
(28)$$\sum_{l=1}^{L}\sum_{m=1}^{M}\sum_{n=1}^{N}X_{jklmn}=0\;,\;\forall \;j,k\in D_{jk}$$

Constraint 28: Physicians tend to be busier on operation days. Hence, to avoid scheduling two tasks in the same period, no outpatient service will be scheduled during surgery periods. 

This hospital and the supported branch hospitals are in the same city, with short distances between them. Therefore, the situation of different allowance for traveling to different hospitals does not occur. However, medical inspection is often postponed because of an excessive number of patients. If different hospitals have consecutive outpatients, the starting time of the next outpatient will be postponed due to traffic problems. Therefore, there will not be consecutive outpatients arranged in different hospitals in the scheduling. Constraints (29)–(33) are to prevent physicians from becoming exhausted due to consecutive services in branch hospitals, which may affect their quality of work.
(29)$$\sum_{m=1}^{M}X_{jk1m1}+X_{jk2m2}\leq 1\;,\;\forall \;j,k$$

Constraint 29: Physicians taking the morning shift in the main hospital cannot continue to take an afternoon shift in a branch hospital.
(30)$$\sum_{m=1}^{M}X_{jk1m2}+X_{jk2m1}\leq 1\;,\;\forall \;j,k$$

Constraint 30: Physicians taking the morning shift in a branch hospital cannot continue to take an afternoon shift in the main hospital.
(31)$$\sum_{m=1}^{M}X_{jk1m3}+X_{jk2m1}\leq 1\;,\;\forall \;j,k$$

Constraint 31: Physicians taking the morning shift in a nursing home cannot continue to take the afternoon shift in the main hospital.
(32)$$\sum_{m=1}^{M}X_{jk1m3}+X_{jk2m2}\leq 1\;,\;\forall \;j,k$$

Constraint 32: Physicians taking the morning shift in a nursing home cannot continue to take the afternoon shift in a branch hospital.
(33)$$\sum_{m=1}^{M}X_{jk2m2}+X_{jk3m1}\leq 1\;,\;\forall \;j,k$$

Constraint 33: Physicians taking the afternoon shift in a branch hospital cannot continue to take the evening shift in the main hospital.
(34)$$\sum_{j=1}^{Na}\sum_{k=1}^{K}\sum_{l=1}^{L}\sum_{m=1}^{M}\sum_{n=2}^{3}X_{jklmn}=Ns1$$

Constraint 34: Total number of senior physicians supporting the branch offices and nursing homes.
(35)$$\sum_{j=Na+1}^{J}\sum_{k=1}^{K}\sum_{l=1}^{L}\sum_{m=1}^{M}\sum_{n=2}^{3}X_{jklmn}=Ns2$$

Constraint 35: Total number of junior physicians supporting the branch offices and nursing homes.

#### 3.3.2. Calculation of the Income and the Relative Weight of Satisfaction

First, this study calculated the maximum income of the hospital and then used the maximum income as a benchmark for physician scheduling to give a relative weight value of 1, while the relative weight of other income was calculated using the maximum income as the denominator, as shown in Equation (36):(36)$$PRW_{n}=\frac{PZ_{n}}{MAX(PZ_{1},PZ_{2}\ldots ,PZ_{n})}$$

Next, within the possible range of the maximum income of the hospital, this study calculated the maximum satisfaction of the physicians with scheduling. Similarly, the maximum satisfaction was used as a benchmark to give a relative weight value of 1, while the relative weight of other types of satisfaction was calculated using the maximum satisfaction as the denominator, as shown in Equation (37):(37)$$SRW_{n}=\frac{SZ_{n}}{MAX(SZ_{1},SZ_{2}\ldots ,SZ_{n})}$$

However, considering that decision makers attach different levels of importance to various scheduling objectives, in order to provide decision makers with more flexible planning, this study took the support income of the hospital as the benchmark and multiplied the relative weight of satisfaction by a decision weight value to show the decision makers’ attention to various scheduling objectives. For example, as shown in Equation (38), the relative weight of satisfaction was multiplied by the sum of the decision weight and the relative weight of income:*DV_n_* = *DW* * P*RW_n_* + (1 − *DW*) * *SRW_n_*(38)

Finally, the maximum decision value could be found through Equation (39) to serve as the basis for management personnel to create schedules:*MDV_n_* = MAX(*DV*_1_, *DV*_2_, …, *DV_n_*)(39)

## 4. Empirical Analysis

This study took the outpatient service status of physicians in a teaching hospital located in southern Taiwan as the research object. The departments of the empirical hospital include internal medicine, obstetrics and gynecology, pediatrics, dermatology, emergency department, etc. We constructed a 0–1 integer programming model based on the number of physicians in the internal medicine department, the outpatient number, and the demand for outpatient visits of the hospital so as to calculate the sum of the maximum relative weight values of the support income of the hospital and the physician satisfaction with arranging an optimal schedule.

### 4.1. Data Summary

In this study, physicians were arranged into appropriate shifts based on the actual outpatient visit demand of the empirical hospital. The shift cycle was one week, excluding Sundays. Eight senior physicians and four junior physicians were, respectively, assigned to three hospitals, i.e., the main hospital, a branch hospital, and a nursing home. Table 1 shows the number of outpatient services of the physicians in one week. For the service staffs being assigned to support external hospitals, the main purposes for the hospital to divide the physicians into senior and junior classes according to the 15-year seniority are: (1) Taiwan medical school focuses on the mentorship system in the learning process. External hospitals give more payments to senior physicians to show their respect. (2) As the public praise of senior physicians may bring more patients to branch hospitals, the supported hospitals are willing to spend more money in inviting senior physicians. The main purpose of supporting the branch hospitals is to help bring more patients. For this reason, seniority becomes the determination criteria for the outpatient in the branch hospitals.

Physicians’ surgery tasks need to be performed during a fixed time. As the physicians’ surgery time could be controlled, and considering the physicians’ fatigue after surgery, they would not be assigned to outpatient services on the date of the surgery, as shown in Table 2. 

### 4.2. Results and Description of the Model

In order to achieve the objectives using the optimal scheduling, it was necessary to execute multiple scheduling programs and calculate the relative weight value of each objective, as shown in Table 3, which shows the schedule of 13 outpatient visits in branch hospitals by eight senior physicians and four junior physicians from the internal medicine department of a medical center. The scheduling scenario concerning each type of hospital income represented the sum of the income obtained from different periods of support by physicians of different types to branch hospitals. Two scenarios can be used as an example. Under scenario 1, the charge for each period of support provided by the senior physicians to branch hospitals is NTD 10,000, while the amount is NTD 5000 in the case of junior physicians. If all 13 periods of outpatient service in branch hospitals are supported by senior physicians, the profits of the main hospital (*PZ*_*1*_) will increase to NTD 130,000. Physician satisfaction (*SZ_n_*) at the right side of Table 3 is the maximum value obtained by the computational formula (1) by setting NS1 of Equation (34) as 13 and *NS_*2*_* of Equation (35) as 0. After obtaining *PZ_n_* and *SZ_n_*, *PRW_n_* and *SRW_n_* are, respectively, obtained according to Equation (37) and Equation (38). Under scenario 2, 12 periods outpatient services in branch hospitals are supported by senior physicians, and one is supported by junior physicians, which will give the main hospital an income of NTD 125,000, and so on. Based on the 13 periods of outpatient services in branch hospitals that require physician support, 14 scheduling scenarios could be made according to the different periods of support by physicians of different seniority.

It takes about 8 h to rearrange the schedule of the internal department by manual scheduling. As shown in Table 3, compared with manual scheduling, the operation by computer using the mathematical model was more efficient, as it was able to find the optimal schedule for both the maximum profit of the hospital and physician satisfaction in one minute and 41 s. Next, the relative weight value of each objective was filled into a Microsoft Excel document in order to calculate different decision-weight values. According to Equation (38), *DV_n_* = *DW* * *PRW_n_* + (1 − *DW*) * *SRW_n_*, managers could adjust the decision weight (DW) according to the practical needs and calculate the decision value of different weights under various scenarios, as shown in Table 4. With senior 2 as the example, when *DW* = 0.7, *DV* = 0.7 * 0.962 + (1 − 0.7) * 0.749 = 0.898, decision makers could then choose the scheduling scenario according to the importance attached to the operating income generated by the provision of support and the physician satisfaction so as to achieve a balance between the two. Taking decision makers’ selection of *DW* = 0.5 as an example, it indicates that they pay equal attention to finance and satisfaction. According to the field of *DW* = 0.5 in Table 5, the decision value under various situations may be obtained, and the data are recorded in Figure 1. As shown in Figure 1, scenario 3 has the best decision value (0.863), which means that when decision makers select equal attention to finance and satisfaction (*DW* = 0.5), the best decision is to assign 11 senior physicians and 2 junior physicians (scenario 3) to branch hospitals for medical inspection. Table 5 is the duty schedule for physicians in the internal medicine department made according to the calculation result of scenario 3. It could be seen from Table 5 that there were no schedules with shifts on three consecutive days, no night shifts in the branch hospital, only morning shifts in the nursing home, and no consecutive shifts in different hospitals. Therefore, the scheduling system developed in this paper could not only calculate the results quickly but could also meet the constraints of the schedule and enable managers to adjust the weight between finances and satisfaction according to the actual needs so as to meet the requirements of hospital management.

### 4.3. Sensitivity Analysis

According to Table 5, the duty quantity of senior and junior physicians in branch hospitals is 11 and 2 times, respectively. Therefore, the weekly support cost PZ3 = charge of senior physicians•11 + charge of junior physicians•2. If the charge of senior physicians each time remains at 10,000 and that of junior physicians is adjusted from −15% to 15%, the variation in the weekly total expenses is shown in Table 6.

If the charge of junior physicians remains unchanged at 5000 but that of senior physicians is adjusted from −15% to 15%; the variation in the weekly total expense is shown in Table 7.

According to Table 6 and Table 7, when the charge of junior physicians is increased to 15%, the total charges only increase by 1.25%; however, when the charge of senior physicians is increased to 15%, the total charges increase by 13.75%. Therefore, it is suggested that under the same satisfaction situation, the expatriate charges of senior physicians may be increased, which will help with the total income of the hospital. However, the burden of the supported hospital should also be considered. If the support cost of junior physicians is increased, hospitals should also increase payments of junior physicians. This will not put much burden onto the supported hospitals; meanwhile, it can increase the junior physicians’ willingness in working in the branch hospitals and their work efficiency.

## 5. Conclusions

In order to expand the business, hospitals tend to assign physicians to other hospitals to help with medical inspections. Besides increasing the line-up of physicians of the supported hospitals, the public praise of physicians from the main hospital will attract more peripheral patients and improve the competitiveness of hospitals. However, expatriate involves physicians’ willingness to work and financial income. This research regarded the income obtained by different expatriates as a weight and takes the number of expatriates as the basis to calculate the score of the best satisfaction as another weight. When decision makers determine the two’s weight ratio according to the actual condition of hospitals, this research may rapidly provide the schedule with the best financial income and the highest satisfaction. This research took a certain hospital in Taiwan and its cooperative hospitals and nursing home as examples. Focusing on decision makers, this system allowed decision makers to determine the weight of finance and satisfaction primarily according to the actual conditions of hospitals. The time spent by administration personnel in making the schedule of physicians is reduced from 8 h to 101 s, thus greatly reducing the burden of administration personnel. In addition, the sensitivity analysis was conducted according to the schedule situation. The analysis results suggested that increasing the expatriate charges of senior physicians may help with the total income. However, this might lead to relatively low willingness in cooperation of the supported hospitals. If the expatriate charge of junior physicians is increased, it may lower the burden of the supported hospitals and increase their willineness in cooperation. This may also enhance the junior physicians’ willingness to work in branch hospitals and their work efficiency.

## Figures and Tables

**Figure 1 healthcare-09-01004-f001:**
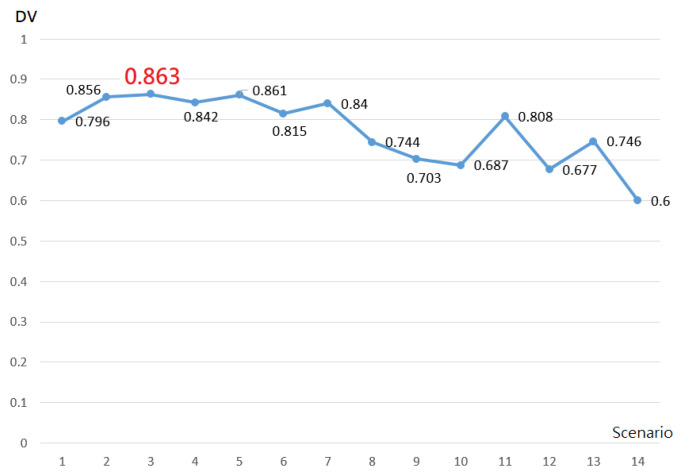
Decision value for each scenario when DW = 0.5.

**Table 1 healthcare-09-01004-t001:** Number of outpatient physicians.

Department	Internal Medicine Department											
Seniority	SP	SP	SP	SP	SP	SP	SP	SP	JP	JP	JP	JP
Code of physician	A1	A2	A3	A4	A5	A6	A7	A8	A9	A10	A11	A12
Shift number of physician	5	5	5	4	4	4	4	4	4	3	3	3
Total shift number of department	48											

Note: SP, senior physicians; JP, junior physicians.

**Table 2 healthcare-09-01004-t002:** Physicians’ surgery schedule.

	Week	MON	TUE	WED	THU	FRI	SAT
Department							
Internal medicine department		A1, A12	A2, A4, A11	A5,A8		A7, A10	

**Table 3 healthcare-09-01004-t003:** Relative weight of each scheduling objective.

Scenarion	Support Income of the Hospital				Physician Satification		Computer Time
	*N* _s1_	*N* _s2_	Profit (*PZ_n_*)	Relative Weight (*PRW_n_*)	Satification (*SZ_n_*)	Relative Weight (*SRW_n_*)	
1	13	0	130,000	1	592	0.592	7 s
2	12	1	125,000	0.962	749	0.749	5 s
3	11	2	120,000	0.923	802	0.802	5 s
4	10	3	115,000	0.885	798	0.798	9 s
5	9	4	110,000	0.846	876	0.876	6 s
6	8	5	105,000	0.808	822	0.822	11 s
7	7	6	100,000	0.769	910	0.91	19 s
8	6	7	95,000	0.731	756	0.756	9 s
9	5	8	90,000	0.692	713	0.713	5 s
10	4	9	85,000	0.654	720	0.72	5 s
11	3	10	80,000	0.615	1000	1	5 s
12	2	11	75,000	0.577	776	0.776	5 s
13	1	12	70,000	0.538	954	0.954	5 s
14	0	13	65,000	0.5	700	0.7	5 s
Total operation time							101 s

**Table 4 healthcare-09-01004-t004:** Decision values (DV) of scheduling scenarios.

Scenario	*PRW_n_*	*SRW_n_*	DW (Decision Weight)										
			1	0.9	0.8	0.7	0.6	0.5	0.4	0.3	0.2	0.1	0
1	1	0.592	**1**	**0.959**	0.918	0.878	0.837	0.796	0.755	0.714	0.674	0.633	0.592
2	0.962	0.749	0.962	0.941	**0.919**	**0.898**	**0.877**	0.856	0.834	0.813	0.792	0.770	0.749
3	0.923	0.802	0.923	0.911	0.899	0.887	0.875	**0.863**	0.850	0.838	0.826	0.814	0.802
4	0.885	0.798	0.885	0.876	0.868	0.859	0.850	0.842	0.833	0.824	0.815	0.807	0.798
5	0.846	0.876	0.846	0.849	0.852	0.855	0.858	0.861	**0.864**	0.867	0.870	0.873	0.876
6	0.808	0.822	0.808	0.809	0.811	0.812	0.814	0.815	0.816	0.818	0.819	0.821	0.822
7	0.769	0.91	0.769	0.783	0.797	0.811	0.825	0.840	0.854	0.868	0.882	0.896	0.91
8	0.731	0.756	0.731	0.734	0.736	0.739	0.741	0.744	0.746	0.749	0.751	0.754	0.756
9	0.692	0.713	0.692	0.694	0.696	0.698	0.700	0.703	0.705	0.707	0.709	0.711	0.713
10	0.654	0.72	0.654	0.661	0.667	0.674	0.680	0.687	0.694	0.700	0.707	0.713	0.72
11	0.615	1	0.615	0.654	0.692	0.731	0.769	0.808	0.846	**0.885**	**0.923**	**0.962**	**1**
12	0.577	0.776	0.577	0.597	0.617	0.637	0.657	0.677	0.696	0.716	0.736	0.756	0.776
13	0.538	0.954	0.538	0.580	0.621	0.663	0.704	0.746	0.788	0.829	0.871	0.912	0.954
14	0.5	0.7	0.5	0.520	0.540	0.560	0.580	0.600	0.620	0.640	0.660	0.680	0.7

**Table 5 healthcare-09-01004-t005:** Physicians’ duty schedule.

	**Monday**						**Tuesday**						**Wednesday**					
	**Main Hospital**				**Branch hospital**	**Nursing home**	**Main Hospital**				**Branch hospital**	**Nursing home**	**Main Hospital**				**Branch hospital**	**Nursing home**
	**OP service 1**	**OP service 2**	**OP service 3**	**Surgery**			**OP service 1**	**OP service 2**	**OP service 3**	**Surgery**			**OP service 1**	**OP service 2**	**OP service 3**	**Surgery**		
MS	A8	A7		A1 A12	A3	A4	A5	A10	A1	A2 A4 A11	A3	A9	A4	A2	A7	A5 A8	A6	
AS					A9		A8				A3		A4	A12	A1		A6	
ES	A3	A10	A11				A1						A1	A2				
	**Thursday**						**Friday**						**Saturday**					
	**Main hospital**				**Branch hospital**	**Nursing home**	**Main hospital**				**Branch hospital**	**Nursing home**	**Main hospital**				**Branch hospital**	**Nursing home**
	**OP service 1**	**OP service 2**	**OP service 3**	**Surgery**			**OP service 1**	**OP service 2**	**OP service 3**	**Surgery**			**OP service 1**	**OP service 2**	**OP service 3**	**Surgery**		
MS			A12		A5	A7	A8		A5	A7 A10	A2		A7	A3	A2			
AS	A10				A6		A5	A4	A9		A2		A11	A6	A1			
ES	A12						A11	A8	A9									

Note: MS, morning shift; AS, afternoon shift; ES, evening shift.

**Table 6 healthcare-09-01004-t006:** Relational table between charge changes of junior physicians and total charges.

Change % for JP	−15%	−10%	−5%	0%	5%	10%	15%
Charge for JP	4250	4500	4750	5000	5250	5500	5750
*PZ* _3_	118,500	119,000	119,500	120,000	120,500	121,000	121,500
Δ*PZ*_3_	−1.25%	−0.83%	−0.42%	0%	0.42%	0.83%	1.25%

**Table 7 healthcare-09-01004-t007:** Relational table between charge changes of senior physicians and total charges.

Change % for SP	−15%	−10%	−5%	0%	5%	10%	15%
Charge for SP	8500	9000	9500	10,000	10,500	11,000	11,500
*PZ* _3_	103,500	109,000	114,500	120,000	125,500	131,000	136,500
Δ*PZ*_3_	−13.75%	−9.17%	−4.58%	0%	4.58%	9.17%	13.75%

## Data Availability

Our data were made available with the submission.

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
