# Peer review of "The Optimization between Physician Satisfaction and Hospital Profit in Cross-Hospital Scheduling—A Case Study of Some Hospitals in Taiwan"

_healthcare, 2021, doi:10.3390/healthcare9081004_

Round 1

Reviewer 1 Report

A language review is required, mostly in the introduction part, and, as well,  I would like the mathematical equation presented with more explanations, in this way the addressability will increase. 

Author Response

Thank you for your constructive comments. Your suggestions have greatly enhanced the quality of our paper. We have revised the paper according to your suggestions.

In this revised manuscript, we highlighted all changes in blue.

Reviewer 2 Report

Manuscript evaluated the satisfactory impact of physician work scheduling vis-à-vis maximizing hospital financial inflows and how both can be optimally integrated to provide better health services and physician satisfaction with their job schedules. Authors adopted a mathematical modeling study design approach and generated formulas, while putting a variety of satisfactory parameters and constraints into consideration. They concluded that their developed system provides better hospital management efficiency and enhanced physician satisfaction.

Manuscript is well written.

  1. Readers are better served if manuscript title includes “…in Taiwan” since manuscript are describing this healthcare system and inferring their results based off on data from it.
  2. The model assumed physician classification solely into senior and junior physicians. What happens when seniority is stratified as senior, middle and junior level physicians? What impact will that have on the developed model.
  3. Also, the charge of support values (NTD 10,000 and 5,000) appear to be selected abstractly. What then happens when charge of support values is modified/changed based on individual hospital cost indices? Will the modifications significantly alter the effectiveness of the current model?
  4. The model gives no consideration to the travel distance or commuting difficulties of physicians to the branch hospital.

Author Response

(The authors gave the same response as above.)

Reviewer 3 Report

The authors have raised a real issue in the medical service industry. I have a few comments for the authors:

(1) discuss and describe other scheduling systems globally to provide better context 

(2) to increase accuracy, the authors can include parameters such as types of duties and etc..

(3) Instead of the seniority of the physician, can the authors also look at the types of physicians in this study? 

(4) to improve the virtual impact & depth of the study, I suggest including a schematic diagram and scenario analysis. 

(5) the conclusion needs to be rewritten with more concise statements and deeper discussion. It is too brief and generic at the moment. 

Author Response

Thank you for your constructive comments. Your suggestions have greatly enhanced the quality of our paper. We have revised the paper according to your suggestions.

In this revised manuscript, we highlighted all changes in purple.

Round 2

Reviewer 2 Report

Authors have made substantial changes to improve the manuscript, based on the prior indicated suggested comments.

Reviewer 3 Report

The paper has improved significantly. The tables and data added has provided better clarity for the readers too. One additional point to highlight to the authors is that, the success of the aviation’s and the police workforce’s scheduling system is also due to their technology integration and communication effort. Can the authors illustrate the implementation of the discussed scheduling system and technology communication between hospitals? One major consideration will be the financial side of things (i.e., payrolls and B2B hospital financial exchange). This is not only a sensitive subject but also a subject that requires absolute accuracy. While this is not the focus of the paper, it will be great for the authors to list the considerations and insights. This will not only be meaningful, but also provide some sense of application feasibility to this study.